# An Overview of Neglected Orthobunyaviruses in Brazil

**DOI:** 10.3390/v14050987

**Published:** 2022-05-07

**Authors:** Helver Gonçalves Dias, Flávia Barreto dos Santos, Alex Pauvolid-Corrêa

**Affiliations:** 1Laboratório de Imunologia Viral, Instituto Oswaldo Cruz, Fundação Oswaldo Cruz (Fiocruz), Rio de Janeiro 21040-900, Brazil; flaviab@ioc.fiocruz.br; 2Department of Veterinary Integrative Biosciences, Texas A&M University, College Station, TX 77843-4458, USA; pauvolid@gmail.com

**Keywords:** arbovirus ecology, transmission cycles, orthobunyaviruses, Brazil

## Abstract

Dozens of orthobunyaviruses have been isolated in Brazil, and at least thirteen have been associated with human disease. The Oropouche virus has received most attention for having caused explosive epidemics with hundreds of thousands of cases in the north region between the 1960sand the 1980s, and since then has been sporadically detected elsewhere in the country. Despite their importance, little is known about their enzootic cycles of transmission, amplifying hosts and vectors, and biotic and abiotic factors involved in spillover events to humans. This overview aims to combine available data of neglected orthobunyaviruses of several serogroups, namely, Anopheles A, Anopheles B, Bunyamwera, California, Capim, Gamboa, Group C, Guama, Simbu and Turlock, in order to evaluate the current knowledge and identify research gaps in their natural transmission cycles in Brazil to ultimately point to the future direction in which orthobunyavirus research should be guided.

## 1. Arthropod-Borne Viruses

Complex networks involving wild vertebrate species, as amplifying hosts, and hematophagous arthropods, as biological vectors, keep the natural maintenance cycles of enzootic arboviruses [1,2,3,4,5]. In general, each group of arboviruses has a specific ecological niche, maintained by a reduced number of species acting as amplifying hosts and biological vectors [4,6].

The co-evolution of certain vertebrate and arthropod species with certain arboviruses has resulted in maintenance cycles sustained by temporary viremia in certain vertebrate species that is high enough to infect hematophagous arthropods that present vectorial competence. Virus isolation or nucleic acid detection of enzootic arboviruses is difficult and fleeting even for amplifying hosts and biological vectors, which impacts diagnosis and, ultimately, surveillance. Reasons include the short viremia in vertebrates, and the small percentage of infected individuals in vector populations [1,4,7,8].

Other species of vertebrates and hematophagous arthropods are also susceptible to infection, but are less likely to play a role in the natural cycles of transmission, and are called terminal hosts and mechanical vectors, respectively [1]. For these, the direct viral diagnosis is even more difficult [1,9]. These peculiar characteristics associated to a non-sensitive surveillance result in an outstanding underreporting of enzootic arbovirus circulation and, consequently, a reduced number of reports of clinical infection where it occurs.

On the other hand, the advances in diagnostic methods have increasingly revealed alternative cycles of transmission involving other vertebrate and invertebrate species, and non-vectorial transmission. Understanding the patterns of arbovirus evolution, adaptation and their interaction with the biological characteristics of vertebrate and invertebrate hosts are key goals for sensitive surveillance programs [10,11].

As the largest Latin-American country, Brazil brings together environmental and sociodemographic aspects that provide for the emergence and establishment of many, most exotic, epidemic arboviruses. Over the last 30 years, the extensive epidemics caused by the four serotypes of dengue virus (DENV) have contributed enormously to the increase in morbidity and mortality statistics in the Americas [12,13,14]. On top of the periodical and explosive epidemics of dengue, in the last decade, the Zika virus (ZIKV) and chikungunya virus (CHIKV) were detected for the first time in Brazil, and since then have been involved in massive epidemics with thousands of fatal cases and permanent neurological impairment [15,16,17].

Brazil is not only highly susceptible to the establishment of urban epidemic arboviruses, but is also considered a fertile ground for the maintenance of enzootic arboviruses due its environmental richness and megadiversity. More than 200 strains of arboviruses have been isolated in Brazil, and from these, over 40 have been reported as pathogenic to humans. The vast majority of these arboviruses have been isolated from the Amazon and are maintained in enzootic cycles of transmission involving species of wildlife, with rodents and birds as amplifying hosts, and species of hematophagous arthropods, such as mosquitoes, midges, and sandflies, as vectors [4,18].

Among the most important enzootic arboviruses reported in Brazil is the yellow fever virus (YFV). Despite being involved in epidemic cycles of transmission in large Brazilian cities in the past, since the 1940s, yellow fever has been restricted to the enzootic cycles of transmission [19]. In the last decade, YFV has been involved in large and recurrent epizootics causing an unprecedented number of hemorrhagic fever outbreaks in non-human primate populations in different regions of the country [19,20,21,22,23].

Three other enzootic arboviruses of increasing importance in Brazil are the Mayaro virus (MAYV, *Togaviridae*), West Nile virus (WNV, *Flaviviridae*) and Oropouche orthobunyavirus (OROV, *Peribunyaviridae*) [24,25,26,27,28,29]. MAYV is an alphavirus involved in epidemics of fever and arthritis in Brazil. MAYV has been increasingly detected in different regions of the country, which includes large metropolitan areas, and for that reason, it also raises concern about potential urbanization [24,25,26]. WNV, which is involved in epidemic and epizooties of neurological disorder in North America and Europe, has been increasingly detected in Brazil. WNV was first detected in western Brazil in 2011, and since then it has been detected in several regions of the country [27,28,29].

OROV is an orthobunyavirus historically involved in explosive outbreaks with hundreds of thousands of cases mostly in the Amazon region. OROV has been sporadically detected in new regions of the country, suggesting that the virus is more spread than originally thought in Brazil [30,31,32].

The emergence and reemergence of arboviruses is a natural phenomenon, but it is greatly influenced by anthropic action [33,34,35,36,37]. Deforestation results in loss of biodiversity, which affects the ecological balance, and this ultimately will allow the occurrence of spillover events to humans and domestic species by pressing the adaptation and exposure to new hosts and maintenance cycles [33,34,35,38]. The rapid and profound environmental degradation underway in Brazil, mostly driven by the opening of agricultural and cattle production frontiers, has greatly affected not only biomes already greatly impacted by agrobusinesses and land-tenure concentration, such as the Cerrado and Amazon Rainforest, but also more recently sensitive biomes until now fairly well preserved, such as the Pantanal wetlands [39,40,41]. The non-stop environmental degradation in Brazil associated with its astonishing biodiversity, which includes hundreds of enzootic viruses, highlights the increased world’s risk for spillover events and consequent emergence of zoonotic viruses, including arboviruses [42].

## 2. Orthobunyaviruses in Brazil

Orthobunyaviruses belong to the newly renamed family *Peribunyaviridae*, order *Bunyavirales* [43]. Virions are mostly spherical, 70–110 nm in diameter, and surrounded by a lipid envelope. The viral particle contains three single-stranded negative sense segments of genomic RNA [44]. Most members of this genus are arthropod-transmitted viruses and for many of them, little is known about ecological aspects of their maintenance cycles. Several orthobunyaviruses have been reported in Brazil and dozens of them are related to human and animal diseases [45,46]. The large majority of these arboviruses were isolated in the Amazon region from mosquitoes and non-human vertebrate hosts [46,47,48,49]. Since the 1950s, novel orthobunyaviruses are being identified and classified by their serological relatedness [48,50,51,52]. There are currently more than 170 known orthobunyaviruses serologically arranged in 18 serogroups. Members of these serogroups are spread around the world. The emergence of the Schmallenberg orthobunyavirus in Europe and Africa, and its association with fetal malformation and abortions in domestic ruminants, resulted in serious economic consequences and triggered the health authorities’ warning regarding neglected orthobunyaviruses [53,54].

At least nine of these serogroups have been reported in Brazil [43,45,46,47,48,49,50,51]. Classical laboratorial techniques used to investigate specific antibodies such as viral neutralization, hemagglutination inhibition assays and complement fixation are still used for classification, surveillance and diagnosis of orthobunyaviruses [51,55,56]. Despite the advance on molecular tools, information regarding genetic properties of several of these viruses remains limited [57]. Reassortment, which is the process whereby related segmented viruses exchange genome segments creating novel reassortant viruses has been well documented among orthobunyaviruses. Based on phylogenetic analysis, there is evidence of reassortment within several orthobunyavirus serogroups including the California, Guama, Bunyamwera and Simbu serogroups [44,58,59,60,61,62].

Of the 57 orthobunyaviruses identified in Brazil, 13 have been isolated from humans (Table 1). Diseases caused by orthobunyaviruses range from self-limited acute febrile illnesses to more severe clinical signs, including hepatitis, aseptic meningitis and encephalitis that can progress to death. Humans develop disease mainly after being bitten mostly by infected mosquitoes [45,56,63].

According to clinical and epidemiological aspects, OROV is the most studied orthobunyavirus in Brazil. Epidemics caused by OROV have been reported since the 1960s mostly in rural areas of the Amazon region [64,65]. However, in recent years, OROV spread has been increasingly evidenced in other regions of the country. In 2016, OROV was detected in saliva and urine samples from patients of Manaus and Salvador, two capitals from the North and Northeast Regions of the country, respectively. OROV has also been increasingly reported in neurological disorder cases in Brazil [32]. These recent findings show that OROV has been silently circulating outside the Amazon region [66,67,68].

## 3. Hosts of Orthobunyaviruses in the Amazon: Deciphering Networks

The Amazon rainforest is the world’s largest ecological reservoir for arboviruses [46,48]. The tropical climate and the great biodiversity of vertebrates and blood-sucking arthropods led to the local establishment, in 1954, of the first virus laboratory of Amazon basin, currently known as the Evandro Chagas Institute, and allowed the isolation of many arboviruses from different phylogenetic groups and with distinct ecological features [48]. Viral isolation has been successful in free-ranging small mammals, including rodents and marsupials, as well as from birds and sentinel animals. Research and recapture programs with vertebrates developed in the 1960s were able to isolate and antigenically characterize dozens of orthobunyaviruses [2,6,56,77].

Rodents and marsupials appear to be hosts of great importance in maintenance cycles of transmission of many orthobunyaviruses. Viral isolation and antibody detection are often successful. Species of the genera *Heteromys, Metachiurus, Nectomys, Oryzomys, Proechimys* and *Zygodontomys* among rodents, and species of *Caluromys, Marmosa* and *Didelphis* among marsupials, can act as amplifying hosts for several orthobunyaviruses [2,46,48,72]. From thirteen orthobunyaviruses of medical importance reported in Brazil, eight (61,5%), including the Apeu orthobunyavirus (APEUV), Caraparu orthobunyavirus (CARV), Itaqui virus (ITQV), Marituba orthobunyavirus (MTBV), Murutucu virus (MURV), Oriboca orthobunyavirus (ORIV), Catu orthobunyavirus (CATUV) and Guama orthobunyavirus (GMAV) are believed to be kept in maintenance cycles involving species of rodents or marsupials [48,72]. Hematophagous arthropod species that have limited flight range not exceeding the average canopy height may have particular importance in transmission cycles involving rodents and marsupials [4,72].

The isolation of the Guaroa orthobunyavirus (GROV), MURV and OROV in bird species, as well as their detection in ornithophilic mosquitoes, such as *Culex* spp. and *Anopheles* spp., suggest that avian hosts may potentially act as amplifying hosts as well in transmission cycles of these orthobunyaviruses in the Amazon [6,47,72] (Figure 1).

## 4. Serogroups of Orthobunyaviruses in Brazil

### 4.1. Anopheles A and Anopheles B Serogroups

The Tacaiuma orthobunyavirus (TCMV) has been associated with acute febrile illnesses in the Amazon region where antibodies are often detected in humans [45]. In São Paulo, southeast Brazil, the TCMV was isolated from a pool of *Anopheles cruzii,* and from a febrile patient during the Rocio virus epidemic in a rural area of a coastal county influenced by the Atlantic Forest biome (Figure 2) [78]. Recently, hemagglutination-inhibition antibodies were found in small ruminants and equids from northeastern Brazil, suggesting that TCMV may circulate in several biomes in Brazil [28]. *Anopheles* species are considered the main vectors of Anopheles A serogroup viruses. However, TCMV has also been isolated from *Aedes triannulatus* and *Haemagogus janthinomys* in Brazil [35]. Regarding possible amplifying hosts, the isolation of TCMV has been restricted to a sentinel Capuchin monkey (*Sapajus apella*) in Brazilian Amazon [46]. The high prevalence of antibodies for TCMV has been found in several wild species of other groups, including bats, small rodents (*Nectomys* and *Oryzomys*) and birds, suggesting that non-primate species may act as amplifying hosts of TCMV in Brazil [6,47,48]. Antibodies for TCMV have been found in horses and buffaloes, indicating these animals have been exposed, and may potentially be used as markers of TCMV circulation in surveillance programs [79,80].

The Arumateua (ARTV), Caraipé (CPEV), Lukuni (LUKV) and Tucuruí (TUCV) viruses are other members of Anopheles A serogroup and have been isolated exclusively from mosquitoes such as *Anopheles (Nyssorhynchus)* sp. and *Aedes* sp. [52]. The only exception is the Trombetas virus (TBTV), which has also been isolated from viscera of the White-lipped peccary (*Tayassui pecari*) [48]. Serosurveys have shown that a wide range of vertebrates has been exposed to Anopheles A serogroup viruses, including wild species of rodents, bats and birds, including Sooty antbird (*Percnostola fortis*) and Blue-crowned manakin (*Lepidothrix coronata*), as well as domestic animals such as horses [6,47,48].

Among the Anopheles B serogroup, the Boraceia virus (BORV) has been isolated from both mosquito species *Anopheles cruzii* and *Wyeomyia pilicauda* in the state of São Paulo. So far, antibodies for BORV have been detected only in humans from the southeast region of the country, and the amplifying hosts of BORV in Brazil remain unknown [81,82].

### 4.2. Bunyamwera Serogroup

The Tucunduba virus (TUCV) has been isolated in Brazil from a single fatal case of a child with acute febrile illness and neurological impairment that presented with meningoencephalitis and progressed to coma and death [45,63]. Clinical infection in non-human vertebrate hosts remains undetected, and species of *Wyeomia* mosquitoes are suspected to be the main vectors. TUCV has also been isolated from other vector species including *Sabethes*, *Psorophora, Limatus, Trichoprosopon* and *Culex* species [47,48,57,69,83]. Amplifying hosts involved in the maintenance cycle of transmission of TUCV remain unknown in Brazil. In a serosurvey conducted with 673 caimans, equines and sheep of the Pantanal, all animals tested negative for TUCV antibodies [30]. The Xingu virus (XINV) has been isolated from a human febrile case diagnosed with hepatitis B that evolved to a fatal outcome [63]. The XINV and Maguari virus (MAGV) are considered highly closely related by some classical tests, but the molecular characterization of XINV confirms them as different orthobunyaviruses [84]. Information about XINV is scarce and its amplifying hosts and vectors, and even cycle of transmission, remain to be discovered. Recently, neutralizing antibodies for XINV were detected in horses and sheep from the Pantanal, west-central Brazil. Equines with monotypic reactions for XINV were detected in two Pantanal ranches. Monotypic reactions for XINV were detected in 14 (6%) of 232 sheep tested in seven out of nine ranches investigated, suggesting widespread circulation of XINV in the region. Among all 14 orthobunyaviruses investigated, XINV presented the highest prevalence among sheep [30].

The Anhembi orthobunyavirus (AMBV) has been isolated from the rodent *Proechimys iheringi* (Ihering’s Atlantic spiny-rat), and from mosquitoes identified with *Wyeomyia* *pilicauda* and *Trichoprosopon pallidiventer* [47,85]. Antibodies of AMBV have been detected in humans in the Amazon [48]. The Kairi orthobunyavirus (KRIV) was isolated from Squirrel monkey (*Saimiri* sp.), from arboreal small rodents of the genus *Oecomys*, and from mosquitoes identified with *Aedes scapularis*, *Psorophora ferox*, *Wyeomyia* spp. and other members of tribe Sabethini [46,48,56,86].

The Macaua orthobunyavirus (MCAV) has been isolated from wild birds, such as common scale-backed antbird (*Willisornis poecilinotus*) and Golden-crowned spadebill (*Platyrinchus coronatus*), and also from rodents such as Guyenne spiny-rat (*Proechimys guyannensis*). Antibodies of MCAV have been detected in humans in the Amazon [48]. Regarding vectors, the only isolation reported occurred in the mosquito species *Sabethes soperi* [6,47,48]. The Taiassui virus (TAIAV) has been isolated from sentinel mice in the Amazon, but its potential amplifying hosts and vectors remain unknown [48].

The Maguari orthobunyavirus (MAGV) was first isolated in Brazil in the 1950s from a mosquito pool containing *Aedes serratus*, *Ae. scapularis*, *Ae. fulvus*., *Anopheles* sp., *Culex taeniopus*, *Mansonia* sp. and *Psorophora ferox* specimens [46,47]. MAGV was previously classified as a subtype of the Cache Valley orthobunyavirus, but some strains of MAGV have been shown to differ antigenically from the prototype. MAGV is now regarded as a closely related, but distinct virus [56,86]. In 1998, MAGV was associated with human disease when was isolated from a patient exhibiting fever, headache, myalgia, and chills in Ucayali, Peru [87]. Antibodies of MAGV have been detected in humans in the Amazon [48], and its enzootic transmission cycle remains unknown. Serological evidence of MAGV exposure has been reported in birds, sheep, water buffalo, man, cattle and mainly horses, from which MAGV has been isolated in Argentina [6,30,79,80,88].

An arbovirus investigation conducted in the Pantanal in the 1990s detected serological evidence for MAGV in 28% of the equines tested [79]. More recently, one sheep and 69 (18.4%) equines from 14 (93%) ranches had monotypic reactions for MAGV in the Pantanal, state of Mato Grosso do Sul, western Brazil [30]. Together, these studies suggest that the circulation of MAGV in equines from the Pantanal has been active for at least three decades and that MAGV is now widely distributed in the region. Among the equines with monotypic reactions for MAGV, three animals were two years old at the moment of the venipuncture in 2009, suggesting that MAGV circulated in equines of the region between 2007 and September 2009 [30]. Additionally, hemagglutination-inhibition antibodies for MAGV were detected in small ruminants and equids by HI from northeastern Brazil [28]. Together, these findings suggest enzootic circulation of MAGV in different regions of Brazil.

Other members of the Bunyamwera serogroup, such as the Cachoeira Porteira orthobunyavirus (CPOV), Iaco virus (IACOV) and Sororoca orthobunyavirus (SORV) have been isolated mainly from acrodendrophilic mosquitoes, including species of *Sabethes*, *Wyeomia* and the tribe Sabethini, respectively [46,47,48,89].

### 4.3. California Serogroup

The Guaroa orthobunyavirus (GROV) has been isolated from human febrile cases, wild passerine as the Pectoral sparrow (*Arremon taciturnus*), and mosquitoes identified with *Anopheles* spp., which are considered the main vectors [6,46,48]. In 2014, in a malaria-endemic rural community close to the city of Iquitos in Peru, GROV infection was detected in 14 febrile persons through a long-term febrile illness surveillance network at local participating health facilities [90]. Antibodies for GROV have also been detected in humans and water buffaloes [80,91,92]. In a serosurvey conducted in the Brazilian Pantanal between 2009 and 2011, neutralizing antibodies for GROV were detected in a sheep that tested seronegative for other 13 orthobunyaviruses. All 375 equines sampled in the same study were seronegative for GROV [30].

Another member of the California serogroup, the Serra do Navio orthobunyavirus (SDNV), has been isolated from *Aedes fulvus* mosquitoes. Antibodies for SDNV have been detected in humans, as well as marsupials and rodents [48,93]. Lastly, the Melao orthobunyavirus (MELV) has been isolated from mosquitoes, including species of *Aedes serratus, Ae. Scapularis* and *Psorophora ferox*, and once from a sentinel non-human primate *Sapajus* [46,48]. In 2014, MELV was isolated from plasma of Haitian children with acute febrile illness during outbreaks caused by alpha- and flaviviruses. Abdominal pain was reported by four case patients with MELV infections, with lymphadenopathy noted in two cases [94].

### 4.4. Capim and Gamboa Serogroups

The members of the Capim serogroup, which includes the Acara orthobunyavirus (ACAV), Benevides orthobunyavirus (BVSV), Benfica virus (BENV), BushBush orthobunyavirus (BSBV), Capim orthobunyavirus (CAPV), Guajara orthobunyavirus (GJAV) and Moriche virus (MORV) have been isolated in the Amazon region from different species of rodents from the genera *Nectomys*, *Oryzomys* and *Proechimys*, as well as from marsupials such as *Caluromys* spp. The main vectors are believed to be species of *Culex*, due to the large number of viral isolations from these mosquitoes [46,47,48,77].

The Gamboa serogroup is represented only by the Gamboa orthobunyavirus (GAMV), which has been isolated in Amazon from wild birds, including *Geotrigon montana* and *Xenops minutus,* known as Ruddy quail-dove and Plain xenops, respectively [6]. Regarding vectors, GAMV has been isolated from *Aedeomyia squamipennis* and *Anopheles triannulatus* mosquitoes [6,95]. None of the members of Capim and Gamboa serogroups have been associated with human disease to date.

### 4.5. Group C Serogroup

Among the genus Orthobunyavirus, the serogroup C has the highest number of viruses associated with human disease [45,63]. The Apeu orthobunyavirus (APEUV), Caraparu orthobunyavirus (CARV), Itaqui virus (ITQV), Marituba orthobunyavirus (MTBV), Murucutu virus (MURV) and Oriboca orthobunyavirus (ORIV) have been isolated from febrile cases in Brazil [56,63,73,74]. A range of vertebrate hosts may be associated with the enzootic transmission of Group C viruses, but species of rodents and marsupials seem to play the main role as amplifying hosts. Most isolations of vertebrate hosts were obtained from rodents, such as species of *Heteromys*, *Nectomys*, *Oryzomys*, *Proechimys* and *Zygodontomys*, and marsupials, such as *Caluromys*, *Didelphis* and *Marmosa*. Antibodies have been detected not only in several other free-ranging wildlife, but also in domestic animals and sentinels (non-human primates and mice) [6,30,47,48,56,71,72,80,96].

CARV is believed to be the most widely distributed Group C virus in the Amazon region. The Itaya virus, a possible reassortant virus containing segment sequences closely related to the CARV and an unidentified group C orthobunyavirus was isolated in 1999 and 2006 from febrile patients in the cities of Iquitos and Yurimaguas in Peru [97]. The detection of antibodies for CARV in humans and animals in the region is frequent. Regarding potential amplifying hosts, CARV has been isolated from Great fruit-eating bats (*Artibeus lituratus*) and rodents, particularly Guyenne spiny rat (*Proechimys guyannensis*) and Large-headed rice rat (*Hylaeamys megacephalus*) [46,48,72]. Nocturnal terrestrial vertebrates and mosquitoes are believed to be amplifying hosts of CARV [98]. Evidence of CARV circulation in Brazil is being mostly reported in the Amazon region [45,48,72,80,91,92,99], but antibodies have been detected in São Paulo, southeastern Brazil [100,101,102]. In a serosurvey conducted in the western Brazil between 2009 and 2011, all 375 equines, 232 sheep and 66 caimans tested were seronegative for CARV [30]. Recently though, hemagglutination-inhibition antibodies for CARV were detected in rural human populations and from several samples from small ruminants and equids by HI from the Northeast Region of Brazil, suggesting that CARV might be more widespread in Brazil than originally thought [28,103].

Although the vast majority of group C viruses has been isolated from rodents, APEUV and MTBV have been isolated only from marsupials, especially Bare-tailed woolly opossum (*Caluromys philander*), Mouse opossum (*Marmosa cinerea*) and Black-eared opossum (*Didelphis marsupialis*) [46,47,48,72]. These findings suggest that transmission occurs in the forest canopy among arboreal marsupials and possibly new-world monkeys and acrodendrophilic vectors [72,98]. Neutralizing antibodies for APEUV were recently reported in domestic animals from West-Central Brazil. Monotypic reactions for APEUV were reported in sheep from three (33%) out of nine ranches sampled in the state of Mato Grosso do Sul. In the same study, hundreds of equines and dozens of free ranging caimans tested negative for APEUV, and all animals sampled in this study tested negative for MTBV [30].

Besides rodents and marsupials, MURV has been eventually isolated from wild birds, such as Long-winged antwren (*Myrmotherula longipennis*) and Cinereous antshrike (*Thamnomanes caesius*), in the Amazon [6]. The loss of native vegetation for agricultural activities has pressured these bird species to adapt to secondary forest and increased contact with humans. This environmental modification may increase the risk of exposure to MURV [98]. There is also a report of MURV isolation from a Pale-throated sloth (*Bradypus tridactylus*) [47]. Little information is available regarding the circulation of Group C arboviruses in Brazil beyond the Amazon. Some recent findings from the Pantanal wetlands located in the southern Amazon, include the detection of neutralizing antibodies for MURV in sheep [30]. According to the data available, *Culex* species seem to play an important role as vectors in transmission cycles of the Group C orthobunyaviruses [46,47,48,72].

### 4.6. Guama Serogroup

Members of the Guama serogroup were first described in the 1960s [56,104], and currently several viruses have been isolated from vertebrates and arthropods in the Amazon region, such as the Ananindeua orthobunyavirus (ANUV), Bertioga orthobunyavirus (BERV), Bimiti orthobunyavirus (BIMV), Cananeia virus (CNAV), Catu orthobunyavirus (CATUV), Guama orthobunyavirus (GMAV), Guaratuba virus (GTRV), Itimirim virus (ITIV), Mirim virus (MIRV), Moju orthobunyavirus (MOJUV) and Timboteua orthobunyavirus (TBTV) [46,48,75,105].

In the Brazilian Amazon region, CATUV and GMAV have been isolated several times from human febrile cases, characterized by mild disease and uneventful recovery [45,63]. CATUV has also been isolated from small rodents (*Nectomys squamipes*, *Oryzomys capito* and *Proechimys guyannensis*), Velvety free-tailed bat (*Molossus obscurus*) and black-eared opossum (*Didelphis marsupialis*) [46,48]. GMAV features a range of hosts, including bats (unidentified species), rodents (*Coendou* sp., *Heteromys anomalus*, *Nectomys squamipes* and *Zygodontomys brevicauda*) and marsupials (*Marmosa murina*, *Didelphis marsupialis*) [4,48,72,105]. Outside the Amazon Basin, little is known about the circulation of GMAV and CATUV. In a recent orthobunyavirus serosurvey conducted in the Pantanal, all equines, sheep and free-ranging caimans tested negative for specific neutralizing antibodies for both arboviruses [30].

ANUV, BIMV, ITIV, MOJUV and TBTV have not been reported causing human disease so far [45,47]. Members of Guama serogroup have been only sporadically isolated from rodents, including species of *Oecomys*, *Oryzomys*, *Nectomys*, *Proechimys*, *Caluromys*, and marsupials such as *Didelphis* [46,48,75]. Avian hosts are not usually involved in transmission cycles of the Guama serogroup members, except GTRV that was isolated from Greenish tyrannulet (*Phyllomyias virescens*) [75]. *Culex* species are considered to be the most important vectors involved in transmission cycles of the Guama serogroup viruses [48,106]. However, some of Guama serogroup orthobunyaviruses have been also isolated from other species of mosquitoes, including *Mansonia*, *Aedes*, *Limatus*, *Psorophora*, *Anopheles*, *Coquilettidia*, *Trichoprosopon* and other genera from the tribe Sabethini [45,48,104]. Besides *Culex* spp., MIRV has also been isolated from *Aedes serratus* and *Psorophora ferox*, and GTRV from *Aedes serratus* and *Anopheles cruzii* [46,48,72,75,106]. In addition to mosquitoes, some Guama serogroup members, such as ANUV, have also been isolated from the midge species *Culicoides paraensis*, which is also the main vector of OROV, and also from sandflies, such as *Lutzomyia* spp. Other members of the Guama serogroup, such as ITIV and TBTV, have never been isolated from hematophagous arthropods [48,72].

### 4.7. Simbu Serogroup

The Simbu serogroup includes only the OROV, Jatobal orthobunyavirus (JATV) and Utinga orthobunyavirus (UTIV). Despite the small number of members, the Simbu serogroup has the greatest clinical and epidemiological importance in Brazil [48]. Just after its first isolation in Brazil from a sloth species *Bradypus tridactylus* captured in a forested area during the construction of the Belém–Brasilia highway, OROV caused several explosive epidemics in the Amazon region. This was likely related to the environmental impact associated with the large concentration of immunologically naïve workers coming from the northeast region of Brazil to the construction zone; in the following year, OROV caused a large epidemic of Oropouche fever in Belém, the capital of the state of Pará, with an estimated 11,000 people affected. With that outbreak, OROV demonstrated its epidemic potential and many other outbreaks have been described subsequently in urban areas in the states of Acre, Amapá, Amazonas, Maranhão, Pará, Rondônia and Tocantins [64,65,107,108,109,110,111,112]. The incubation period of OROV ranges from three to eight days and can result in sudden acute febrile illness accompanied in most cases by headache, myalgia, arthralgia, exanthema and photophobia [63,113]. Clinical recurrence of one or more symptoms are observed in a proportion of patients during epidemics [113,114]. Although there are sporadic reports of cases of oropouche fever with neurological complications, data of incidence during epidemics are unknown [32,115]. Among the reports of neurological disease, cases with meningitis or meningoencephalitis with detection of OROV RNA ou IgM in cerebrospinal fluid are the most frequent [32,115,116]. Immunocompromised patients seem to be the most affected, but cases of oropouche fever have been reported also in healthy individuals [32,117]. Although several viruses of the Simbu serogroup have been incriminated as pathogenic for domestic and wild animals, including clinical evidence of a teratogenic infection, this effect has not been commonly linked to OROV infection [118,119,120].

In the last decade, evidence of OROV has been more often reported outside the Amazon region. Recent reports indicate that OROV circulates also in the west-central [30,66], northeast and southeast regions of the country. Autochthonous cases of oropouche fever, as well antibodies detection, were recently reported in Bahia state, which is of concern as the state located in northeast Brazil is famous for attracting hundreds of thousands of tourists from all over the world every year for Carnival celebrations [67,103].

In the southeast region, which is the most populated region of Brazil, cases of infected individuals returning from the Amazon and forested areas from the northeast have been recorded in the state of São Paulo [117,121]. The silent circulation of OROV in forested areas of different regions of Brazil associated with the wide geographic distribution of its main vector *Culicoides paraensis* and other potential vectors as *Culex* spp. in the country is of concern for the potential urbanization of oropouche fever in Brazil [66,122,123,124]. Despite the epidemic potential, there are very few studies on the prevalence of oropouche fever. Investigation of OROV should be opportunely incorporated in arbovirus surveillance systems throughout the country [67,125,126]. The main amplifying species of vertebrates involved in OROV cycles of transmission remain to be discovered. Besides the first isolation from sloths, OROV has also been isolated from birds and other mammals [6,48]. There are reports of OROV isolation from columbiformes such as the ruddy ground dove (*Columbina talpacoti*) in the Brazilian Amazon between 1960 and 1980 [6], and from marmosets such as *Callithrix* sp. collected in 2000 from the state of Minas Gerais, southeast Brazil [76]. Antibodies have been detected in a much larger number of bird and mammal species, including New World monkeys such as the *Alouatta* and *Sapajus* species, and different bird species [6,127,128]. Among the bird families reported with OROV antibodies are Dendrocolaptidae, Fringillidae, Furnariidae, Momotidae, Formicariidae, Tyrannidae, Fringillidae, Galbulidae, Thraupidae, Vireonidae, Cotingidae, Troglodytidae and Pipridae [6]. The exposure to OROV has also been reported in domestic species as sheep and water buffalo [30,80].

During epidemics, OROV is frequently detected in midges *Culicoides paraensis*, which is considered the most important vector. *Culicoides paraensis* is easily found infesting banana plantations in Brazil, where optimal environmental conditions for vector reproduction are found [114,122]. Populations of *Culicoides paraensis* are widespread throughout Latin America and can also be found in southeastern Brazil [122,123]. OROV has also being detected from culicids such as *Aedes serratus*, *Ae. aegypti* and *Culex quinquefasciatus* in the Amazon and west-central region of Brazil. The participation of *Culex quinquefasciatus* as vector of OROV during epidemics is being speculated, but the role of this species as primary vector remains unclear [29,110,129].

Two other members of the Simbu group are JATV, which was first isolated from the procyonid South American Coati *Nasua nasua* in 1985, and UTIV, which was isolated from sloth *Bradypus tridactylus* in 1965, both in the state of Pará, North Region of Brazil [130,131]. None of these viruses have been reported causing human disease so far. Vectors remain unknown since both orthobunyaviruses have never been isolated from hematophagous arthropods [48]. For these arboviruses without known vectors, the permissiveness in mosquito cell cultures and the antigenic and genetic characteristics, which should be similar to those of known arboviruses, are taken into account [132]. Antibodies for UTIV have been detected in humans, wild birds (Furnariidae, Pipridae and Formicariidae), domestic pigs, bat, marsupials, rodents, monkeys, water buffalo and edentates [6,46,47,80].

Regarding JATV, on the basis of nucleotide sequencing, the small (S) segment of RNA is very similar to the Peruvian genotype of OROV suggesting that JATV is a reassortant containing the S RNA of OROV and other non-identified viruses of Simbu serogroup [131]. Genetic reassortment between viruses of the same serogroup is a common natural phenomenon and allows the emergence of new strains [62]. The Iquitos virus and Madre de Dios virus have been isolated from human febrile cases in Peru [61]. Another new recombinant OROV has been isolated from *Callithrix* sp. in Brazil in 2015 and named the Perdões virus (PERDV) [133].

### 4.8. Turlock Serogroup

The Turlock orthobunyavirus (TURV) has been often isolated in the Amazon region from different species of wild birds, including the species of the genera *Chiroxiphia, Geotrygon*, *Hylophylax*, *Myrmoborus*, *Pyriglena* and *Thamnophilus* [6,134]. TURV has also been isolated from sentinel mice, but the viral isolation and detection of antibodies in a large number of bird species suggest that avian hosts, rather than rodents, participate as amplifying hosts of TURV in the Amazon region [46]. Despite isolation from mosquitoes identified with *Wyeomyia* species, *Culex portesi* has been suggested one of the potential vectors of TURV [48,69].

## 5. Ungrouped Orthobunyaviruses Isolated in Brazil

Some other orthobunyaviruses with unrecognized medical importance and very little information are still to be classified according to physicochemical properties, genetic and antigenic relationships. The Belém virus (BLMV) has been isolated from wild birds, such as spot-backed antbird (*Hylophylax naevius*) and East Amazonian fire-eye (*Pyriglena leuconota*), in the Brazilian Amazon, and antibodies have been detected in humans, birds and water buffaloes [6]. No vectors of BLMV have been identified yet.

The Enseada orthobunyavirus (ENSV) has been isolated from mosquitoes *Culex epanastasi* and *Culex taeniopus* mosquitoes, and the vertebrate hosts involved in the maintenance in nature remains unknown [75,135]. The Mojuí dos Campos virus (MDCV) has been isolated from an unidentified bat and antibodies have been detected in wild birds [6,47]. The vectors involved in the transmission of MDCV remain to be discovered. The Pacora-like virus (PACV-like) has been isolated from wild bird (*Automolus ochrolaemus*) and *Culex* mosquitoes [6,136]. The Pará virus (PARAV) has only been isolated from sentinel mice, and natural hosts and vectors remains unknown. The Santarém virus has been isolated from rodents *Oryzomys* sp. and antibodies have been detected in *Proechimys* sp. So far, sandflies *Lutzomyia carrerai* is the only species from which the Santarem virus has been isolated [47,48].

## 6. Ecological Gaps in Orthobunyavirus Maintenance Cycles in Brazil

Emergence, re-emergence and spillover mechanisms of neglected enzootic orthobunyaviruses can be better understood from the knowledge of the ecological niches that each one occupies. For many of the orthobunyaviruses described so far, there is little or no information about transmission cycles [48]. Many ecological aspects remain unknown, mainly about amplifying hosts, arthropod vectors and ecological niches. Understanding the ecological dynamics of each of these viruses is crucial for an efficient response to ultimately mitigate possible outbreaks and/or epizootics [6].

At least thirteen orthobunyavirus (Anopheles A, Bunyamwera, California, Group C, Guama and Simbu serogroups) have been reported to be able to cause disease in humans [63]. Increased travelling increases also the risk for transmission of zoonotic orthobunyaviruses to non-endemic areas. From what has been exposed above, it is clear that the prevalence of orthobunyavirus in Brazil is highly underestimated.

From the identification of the main vector species and vertebrate amplifying hosts in enzootic transmission cycles, it is possible to recognize knowledge gaps and point to the future direction in which scientific research should be guided. In the light of the current coronavirus pandemic originated from a spillover event that resulted in millions of deaths and unprecedent economic losses worldwide, these gaps constitute a wake-up call for authorities, academia, and funders to enhance research in orthobunyavirus diseases in Brazil.

## 7. Lack of Tools and Protocols for the Diagnosis of Orthobunyaviruses

Many ecological aspects of the transmission and maintenance cycles of orthobunyaviruses remain unknown and the reduced information is greatly influenced by the vicious cycle involving reduced investigation and limited diagnostic tools. Clinical diagnosis is hampered by mostly mild illnesses and nonspecific clinical signs and symptoms [63]. As most of these viruses have been reported in forested areas mainly from the Amazon region, riverside and indigenous populations are possibly the most exposed and infected [45,49,109,137,138]. The difficult access to health services in isolated areas of the Amazon also contributes to the lack of knowledge regarding incidence in the human population [45]. Except for OROV, orthobunyaviruses have not been related to epidemics [48]. The silent circulation of these viruses results in great underestimation of clinical infection that results in scarce public investment and consequent non-sensitive surveillance and neglection [48,111,113].

Another challenge for the detection and better understanding of the ecological aspects involved in the transmission of these arboviruses is the diagnostic. Very few of these viruses have molecular diagnostic tools available, which makes rapid and accurate detection difficult [139]. Because of their obscure occurrence, there are still no commercial diagnostic kits, whether serological or molecular, for detecting most of the orthobunyaviruses discussed here [139,140]. The absence of appropriate tools for the identification of these viruses aggravates their sub-notification. Therefore, it is urgent to develop rapid methodologies, such as ELISA-based tests or RT-PCR, for use in diagnosis and in epidemiological investigations [139,141].

In Brazil, few institutions are able to carry out specific diagnosis of these neglected agents, which is mostly restricted to laborious and specific serological methods such as neutralization tests and advanced molecular methods such as metagenomics and nucleotide sequencing. Orthobunyaviruses present a scarcity of data in genetic information databases, which makes the development of molecular tests even more complex and difficult [140]. Both approaches remain restricted to very few reference and university laboratories [30,141]. The restriction of diagnosis contributes enormously to a less sensitive surveillance. The Amazon region harbors Brazil’s National Reference Laboratory for Arbovirus Diagnosis, which is responsible for most of the discoveries regarding the orthobunyaviruses in Brazil [18,48]. With the increasing detection of orthobunyaviruses outside the Amazon, efforts should be made to increase surveillance capacity nationwide [125,126].

As transmission of orthobunyaviruses is mainly enzootic, and domestic animals and humans may develop a short period of viremia [2,45,72], the detection of active infection with molecular methods and subsequent nucleotide sequencing is challenging. Long before thinking about using modern metagenomics and next generation sequencing, arbovirologists were able to identify and categorize different groups of orthobunyaviruses based only in vivo models using mostly mice and cell cultures [132,141]. Cell lines, mainly VERO (kidney epithelial cells from African green monkey) and C6/36 (derived from larvae of *Aedes albopictus*) are highly susceptible to most arboviruses and therefore remain as very important models for viral isolation, identification and prevalence in arbovirus diagnostic [132]. Detection of antibodies remains as one of the most valuable and informative approaches for the diagnostic of these sylvatic arboviruses. Hemagglutination-inhibition (HI) has been used as diagnostic method for the detection of arbovirus group-specific antibodies in the Amazon [132,142]. Despite lower specificity when compared to neutralization assays, the advantage of the HI is the capacity to concomitantly investigate antibodies for antigens of several groups of arboviruses. Because of that, the use of HI is particularly informative to assess the local to different arbovirus groups [28]. For the reason, HI is still used as a screening test for more specific serological tests designed to detect neutralizing antibodies [28].

Other serological tools commonly used for the investigation and diagnostic of enzootic arboviruses are the virus neutralization test (VNT) or plaque reduction neutralization test (PRNT), and complement fixation (CF). These assays may prove useful for the detection of closely related viruses, such as viruses belonging to the same serogroup [30,55]. FC has greater specificity when compared to HI, but requires specifically trained personnel [132]. The VNT or PRNT are considered gold standards for confirmation of serological test results, as they are sensitive and present high specificity allowing the differentiation among related viruses [30,143]. However, because VNT and PRNT require the use of cell cultures and infectious viruses to be manipulated by highly trained personnel in high-level biosafety facilities, these methods are considered laborious, expensive and time-consuming for large scale testing. For this reason, VNT and PRNT have been used primarily as confirmatory tests [28,30,143]. The great challenge for serology for orthobunyaviruses in Brazil relies on the concomitant circulation of several antigenically related viruses, which increases cases of false positive results. The use of the IgM Antibody Capture Enzyme-Linked Immunosorbent Assay (MAC-ELISA), which has betterspecificity when compared to other screening methods, could also be an option, but it would need further evaluations among cross-reactivity in the different serogroups. Considering the technical limitations of the above-mentioned methods, a surrogate screening test for the serological investigation of orthobunyaviruses in Brazil must be continuously pursued. The suitable screening method would ideally be a fast and qualitative kit for the detection of specific neutralizing antibodies on competitive ELISA or chromatographic immunoassay platforms. Few options with a similar approach are already available for the investigation of flaviviruses, including epitope-blocking ELISA and the VecTest-inhibition assay, and merit further evaluation [143].

Another obstacle for the use of serology for the investigation of orthobunyaviruses is their capacity for reassortment. This phenomenon can have significant implications due to the possibility of the emergence of a virus with increased pathogenicity [59]. Reassortment of genomic RNA segments involving OROV has been reported in isolates obtained in countries of South America [61,62,133].

## 8. Conclusions and Future Directions

Orthobunyaviruses represent an important and neglected group of arboviruses in Brazil. Human infections range from mild clinical signs to rare neurological events. According to the data collected, mosquitoes of the genus *Culex* play a central role in the transmission of most serogroups, but other mosquitoes, such *Psorophora* spp., *Mansonia* spp., *Anopheles* spp., *Wyeomyia* spp., *Sabethes* spp. and *Aedes* spp. [46,48,72], as well as other non-mosquito vectors, such as midges, are associated with orthobunyavirus transmission in Brazil [114,129].

Although multiple different orthobunyaviruses have been described in vertebrate hosts in Brazil, most are not associated with veterinary or human disease. Brazil is home to one of the largest vertebrate biodiversity in the world, including non-human primates, rodents, marsupials and bats, in addition to hundreds of mosquito species. Different characteristics combined contribute to Brazil being a hotspot for the emergence of new zoonotic viruses, which includes arboviruses with epidemic potential [11]. Despite increasing research and surveillance sensitivity for arboviruses in Brazil over the past five decades, continuous syndromic and vector monitoring programs are instrumental and should be encouraged, especially those focusing on the most affected areas. Effective prevention and control measures, associated with active surveillance based on one health approach including the participation of not only public health, but also environmental authorities and members of civil society, must be guided and implemented. A sustainable exploration of natural resources is instrumental and must also be strengthened as a measure to prevent and mitigate spillover events and the emergence of agents with zoonotic potential.

## Figures and Tables

**Figure 1 viruses-14-00987-f001:**
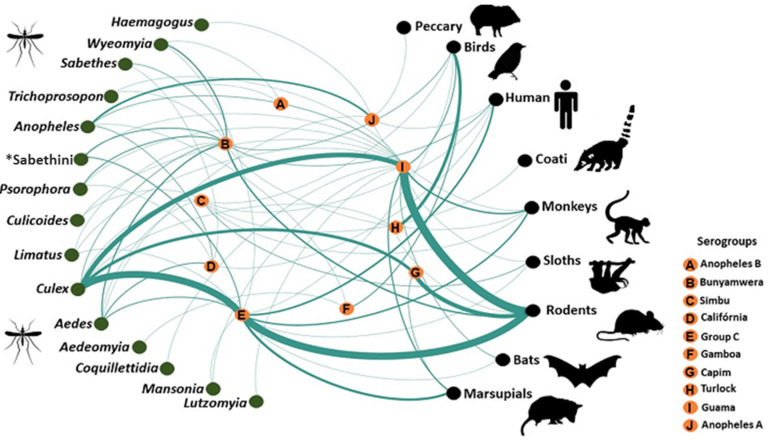
Network of orthobunyaviruses isolated in Brazil by serogroups, hosts and vectors. Lines and their thickness represent number of hosts shared between viruses. Each green dot (left side) represents vector genera; the black dots (right side) represent vertebrate host groups; in the center of the network, represented by orange dots, are the orthobunyavirus serogroups related to both vectors and vertebrate hosts. A presence/absence matrix was built to show the distribution of the viruses according to the vectors and vertebrate hosts associated with them. Network analyses and visualization were performed on the platform Gephi (https://gephi.org; acessed on 1 August 2021), using the force-directed algorithm ForceAtlas2, followed by local rearrangement for visual clarity, leaving the network’s overall layout unperturbed. Tribe Sabethini includes 435 currently recognized species that comprise 14 genera. Older studies found great difficulty in characterizing Sabethini adults based on morphometric patterns. The data used to include interactions with the tribe Sabethini refer only to this taxonomic level, since there was no more precise information about the genus or even the species.

**Figure 2 viruses-14-00987-f002:**
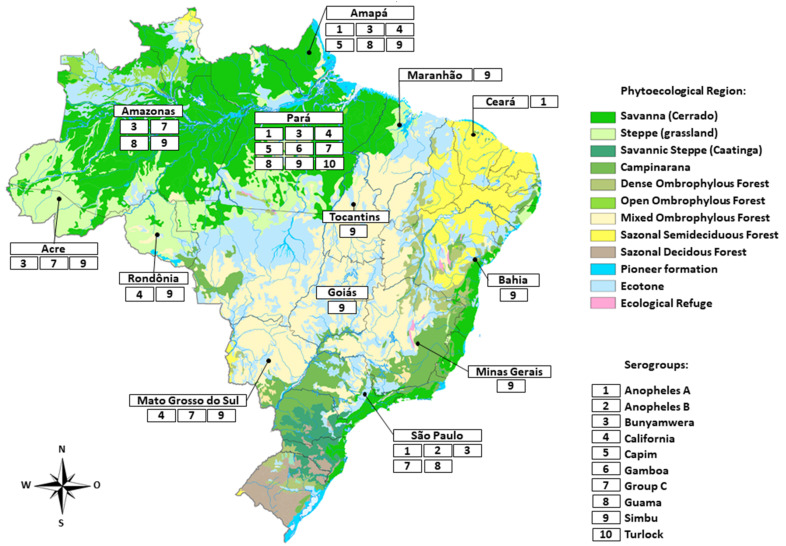
Map of Brazil showing states where orthobuniaviruses of several serogroups have been reported.

**Table 1 viruses-14-00987-t001:** Summary of human pathogenic orthobunyaviruses isolated in Brazil.

Serogroup	Virus	Human	Wildlife Species	Sentinel Animals	Invertebrates	References
Anopheles A	Tacaiuma orthobunyavirus (TCMV)	Yes	Unreported	*Sapajus apella* (Capuchin monkeys)	*Aedes triannulatus, Anopheles cruzii, Haemagogus janthinomys*	[4,35,45,46,47,56,63,69]
Bunyamwera	Tucunduba virus (TUCV)	Yes	Unreported	Unreported	*Anopheles* (Nys.) sp., *Culex coronator, Limatus flavisetosus, Li. durhanii, Psorophora ferox, Wyeomyia* sp.	[4,45,57,63,69]
Bunyamwera	Xingu virus (XINV)	Yes	Unreported	Unreported	Unreported	[4,45,48,63]
California	Guaroa orthobunyavirus (GROV)	Yes	Unreported	*Arremon taciturnus* (Pectoral sparrow)	*Anopheles nunestovari, An triannulatus*	[6,45,47,70]
Group C	Apeu orthobunyavirus (APEUV)	Yes	*Caluromys philander* (Bare-tailed woolly opossum), *Marmosa cinerea* (Robinson’s mouse opossum)	*Sapajus apella* (Capuchin monkeys), mice	*Aedes arborealis, Ae. septemstriatus, Culex aikenii, Culex* sp.	[45,47,56,71,72]
Group C	Caraparu orthobunyavirus (CARV)	Yes	*Artibeus lituratus* (Great fruit-eating bat)*, Heteromys anomalus* (Trinidad spiny pocket mouse)*, Nectomys squamipes* (Scaly-footed water rat)*, Oryzomys laticeps* (Large-headed rice rat), *Proechimys guyannensis* (Guyenne spiny rat), *Zygodontomys brevicauda* (Short-tailed zygodont)	Monkey (unreported specie)	*Aedes scapularis*, *Ae. serratus*, *Culex aikenii*, *Cx. sacchettae, Cx. portesi*, *Cx. vomerifer*, *Cx. spissipes*, *Cx. coronator, Cx. nigripalpus, Cx. accelerans*, *Cx. amazonensis*, *Limatus durhamii*, *Wyeomyia medioalbipes*, Sabethini	[45,47,56,72,73]
Group C	Itaqui virus (ITQV)	Yes	*Marmosa murina* (Linnaeus’s mouse opossum), *Metachirus nudicaudatus* (Brown four-eyed opossum), *Nectomys squamipes* (Scaly-footed water rat)*, Oryzomys capito* (Large-headed rice rat)*, Proechimys guyannensis* (Guyenne spiny-rat)	*Sapajus* (Capuchin monkey), hamster and mice	*Culex aikenii, Cx. portesi, Cx. vomerifer, Cx. spissipes*	[45,47,72,74]
Group C	Marituba orthobunyavirus (MTBV)	Yes	*Caluromys philander* (Bare-tailed woolly opossum), *Didelphis marsupialis* (Black-eared opossum)	*Sapajus* (Capuchin monkey) and mice	*Culex aikenii, Cx. portesi*	[45,46,47,72]
Group C	Murutucu virus (MURV)	Yes	*Bradypus tridactylus* (Pale-throated sloth), *Didelphis marsupialis* (Black-eared opossum), *Marmosa cinerea* (Robinson’s mouse opossum), *Myrmotherula longipennis* (Long-winged antwren), *Nectomys squamipes* (Scaly-footed water rat), *Oryzomys capito* (Large-headed rice rat), *Proechimys guyannensis* (Guyenne spiny rat), *Thamnomanes caesius* (Cinereous antshrik)	*Sapajus* (Capuchin monkey) and mice	*Culex aikenii, Cx. portesi, Cx. vomerifer,* Sabethini.	[6,45,46,47,48,56,72]
Group C	Oriboca orthobunyavirus (ORIV)	Yes	*Didelphis marsupialis* (Black-eared opossum)*, Marmosa cinerea* (Robinson’s mouse opossum), *Nectomys squamipes* (Scaly-footed water rat)*, Oryzomys capito* (Large-headed rice rat), *Proechimys guyannensis* (Guyenne spiny rat)	*Sapajus* (Capuchin monkey) and mice	*Aedes* spp.*, Culex portesi, Mansonia* spp.*, Psorophora ferox,* Sabethini	[47,48,56,72]
Guama	Catu orthobunyavirus (CATUV)	Yes	*Didelphis marsupialis* (Black-eared opossum), *Molossus obscurus* (Velvety free-tailed bat), *Nectomys squamipes, Oryzomys capito* (Large-headed rice rat), *Proechimys guyannensis* (Guyenne spiny rat)	*Sapajus* (Capuchin monkey) and mice	*Anopheles nimbus, Culex portesi, Culex* spp.	[4,45,46,47,56,72,75]
Guama	Guama orthobunyavirus (GMAV)	Yes	Bat (unidentified species)*, Coendou* sp. (Prehensile-tailed porcupine), *Heteromys anomalus* (Trinidad spiny pocket mouse), *Nectomys squamipes* (Scaly-footed water rat), *Zygodontomys brevicauda* (Short-tailed zygodont)	*Alouatta* (Howler monkey), *Sapajus* (Capuchin monkey), hamster, mice and *Oryzomys* (Marsh rice rat)	*Culex portesi, Cx. vomerifer, Cx. taeniopus, Lutzomyia* sp.*, Mansonia* spp.*, Limatus* spp.*, Sabethes chloropterus, Psorophora* sp.*, Trichoprosopon* sp.	[4,45,46,47,56,63,69,72,75]
Simbu	Oropouche orthobunyavirus (OROV)	Yes	*Bradypus tridactylus* (Pale-throated sloth), *Callithrix* sp. (marmoset), *Columbina talpacoti* (Ruddy ground dove)	Unreported	*Aedes serratus, Culicoides paraensis, Culex quinquefasciatus*	[46,47,48,64,76]

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
