# Peer review of "An Overview of Neglected Orthobunyaviruses in Brazil"

_viruses, 2022, doi:10.3390/v14050987_

Round 1
Reviewer 1 Report
In this manuscript, Helver Gonçalves Dias et al gave a highly informative overview of neglected orthobunyaviruses in Brazil. The authors described the important aspects regarding their enzootic cycles of transmission, amplifying hosts and vectors, and biotic and abiotic factors involved in spill-over events to humans. Overall, this was an interesting review and a well-organized manuscript that had offered a large amount of data that are very informative.
I have no serious concerns with the methodologies or any major issues with the manuscript in general. Only some minor points need to be addressed:
- In the first part--Arthropod-borne viruses, the authors might shorten the length of the introduction and focused on the background of orthobunyavirus, while give only a brief introduction of others than orthobunyavirus.
- Table 1. The authors might provide a brief description of the clinical features of human infection with these pathogenic orthobunyaviruses isolated in Brazil.
- The first report time of the serogroups of the orthobunyaviruses should be incorporated in a way to the manuscript. There are really useful information for the comprehension of the virus.
- The Abstract might be strengthened to reflect the major points of the review. In my opinion, the main serotypes that have been reviewed might be listed.
Author Response
Point 1: In the first part - Arthropod-borne viruses, the authors might shorten the length of the introduction and focused on the background of orthobunyavirus, while give only a brief introduction of others than orthobunyavirus.
Response: Thanks for your suggestion. We revised the manuscript accordingly.
Point 2: Table 1. The authors might provide a brief description of the clinical features of human infection with these pathogenic orthobunyaviruses isolated in Brazil.
Response: Thank you for your suggestion. Unfortunately, information about the clinical features caused by orthobunyaviruses in Brazil is scarce. With few exceptions, most reports are restricted to the description of acute febrile illness (rapid onset of fever and other symptoms). Because of that, when available, we described clinical features of each serogroup in their respective sections throughout the text.
Point 3: The first report time of the serogroups of the orthobunyaviruses should be incorporated in a way to the manuscript. There are really useful information for the comprehension of the virus.
Response: Thanks for this suggestion. Most of these viruses were first isolated in the 1960s to 1980s and some reports have limited information available. To make it even more difficult, some of these descriptions have not been officially published and are only listed as 'personal communication'. Whenever possible, we cite the first report time of the serogroups.
Point 4: The Abstract might be strengthened to reflect the major points of the review. In my opinion, the main serotypes that have been reviewed might be listed.
Response: We agree that is useful information for the absract, and revised the manuscript accordingly.
Reviewer 2 Report
This paper reviews the state of knowledge regarding orthobunyaviruses in Brazil. Orthobunyaviruses are an important and neglected group of arboviruses, and Brazil has historically had comparitively more activity (or identification thereof) related to these pathogens in South America. This is a nicely written review of arbovirus activity in Brazil. Further, it provides excellent evidence of the lack of preparedness and need for basic diagnostic tools for these arboviruses.
Major comment:
There is no discussion regarding the effort of detecting and surveilling for arboviruses in Brazil relative to other countries. This is an important distinction as Brazil has competitively performed more research and surveillance into these viruses. I think it is worth mentioning this to put the situation in Brazil into the appropriate regional and global context.
Minor comments:
Lines 65-66: “In the last decade, YFV has been involved in large and recurrent epizootics affecting greatly non-human primate populations in different regions of the country.” It is unclear what “affecting greatly non-human primate populations” means explicitly.
Line 109 (and throughout): Schmallengberg orthobunyavirus is italicized while throughout other viruses are not. What is the current convention? Needs to be standardized.
Line 236: there’s a font difference (size?)
Author Response
Point 1: There is no discussion regarding the effort of detecting and surveilling for arboviruses in Brazil relative to other countries. This is an important distinction as Brazil has competitively performed more research and surveillance into these viruses. I think it is worth mentioning this to put the situation in Brazil into the appropriate regional and global context.
Response: Thanks for your suggestion. We revised the manuscript accordingly.
Point 2: Lines 65-66: “In the last decade, YFV has been involved in large and recurrent epizootics affecting greatly non-human primate populations in different regions of the country.” It is unclear what “affecting greatly non-human primate populations” means explicitly.
Response: Thanks for your suggestion. We have rephrased accordingly.
Point 3: Line 109 (and throughout): Schmallengberg orthobunyavirus is italicized while throughout other viruses are not. What is the current convention? Needs to be standardized.
Response: Thanks for your comment. We revised the manuscript accordingly.
Point 4: Line 236: there’s a font difference (size?)
Response: Thanks for your comment. We revised the manuscript accordingly.